# Predictive Value of Estimated Lean Body Mass for Neurological Outcomes after Out-of-Hospital Cardiac Arrest [note 1]

**DOI:** 10.3390/jcm10010071

**Published:** 2020-12-28

**Authors:** Sung Eun Lee, Hyuk Hoon Kim, Minjung Kathy Chae, Eun Jung Park, Sangchun Choi

**Affiliations:** 1Department of Emergency Medicine, School of Medicine, Ajou University, 164 Worldcup-ro, Yeongtong-gu, Suwon 16499, Korea; plumpboy@hanmail.net (S.E.L.); mutjeo@gmail.com (M.K.C.); amita62@nate.com (E.J.P.); avenue5933@gmail.com (S.C.); 2Department of Neurology, School of Medicine, Ajou University, Suwon 16499, Korea

**Keywords:** lean body mass, out-of-hospital cardiac arrest, prognosis, neurological outcome, neuroprotective effect

## Abstract

Background: Postcardiac arrest patients with a return of spontaneous circulation (ROSC) are critically ill, and high body mass index (BMI) is ascertained to be associated with good prognosis in patients with a critically ill condition. However, the exact mechanism has been unknown. To assess the effectiveness of skeletal muscles in reducing neuronal injury after the initial damage owing to cardiac arrest, we investigated the relationship between estimated lean body mass (LBM) and the prognosis of postcardiac arrest patients. Methods: This retrospective cohort study included adult patients with ROSC after out-of-hospital cardiac arrest from January 2015 to March 2020. The enrolled patients were allocated into good- and poor-outcome groups (cerebral performance category (CPC) scores 1–2 and 3–5, respectively). Estimated LBM was categorized into quartiles. Multivariate regression models were used to evaluate the association between LBM and a good CPC score. The area under the receiver operating characteristic curve (AUROC) was assessed. Results: In total, 155 patients were analyzed (CPC score 1–2 vs. 3–5, *n* = 70 vs. *n* = 85). Patients’ age, first monitored rhythm, no-flow time, presumed cause of arrest, BMI, and LBM were different (*p* < 0.05). Fourth-quartile LBM (≥48.98 kg) was associated with good neurological outcome of postcardiac arrest patients (odds ratio = 4.81, 95% confidence interval (CI), 1.10–25.55, *p* = 0.04). Initial high LBM was also a predictor of good neurological outcomes (AUROC of multivariate regression model including LBM: 0.918). Conclusions: Initial LBM above 48.98kg is a feasible prognostic factor for good neurological outcomes in postcardiac arrest patients.

## 1. Background

Patients with return of spontaneous circulation (ROSC) after out-of-hospital cardiac arrest have low survival rates and poor neurological prognosis despite active treatments, such as intensive care unit (ICU) care and targeted temperature management (TTM) [1]. Critically ill patients with hypoxic–ischemic insults during cardiac arrest may be further injured by myocardial dysfunction, impaired cerebral autoregulation, and systemic inflammatory reaction after ROSC. Consequently, patients are at a risk of multiorgan damage, including the nervous system damage [2,3,4].

Although obesity, which is defined as a body mass index (BMI) ≥ 30 kg/m^2^, is considered an important risk factor for cardio-cerebrovascular diseases, metabolic syndrome, overweight, and obese patients have better prognosis in critical conditions, thereby creating an “obesity paradox” [5,6,7]. Improved survival rates and better neurological outcomes were observed in postcardiac arrest patients with relatively higher BMI [8,9,10]. Although mechanisms underlying this “obesity paradox” have not been clearly elucidated, the protective effect of obesity is based on large metabolic storage, less cachexia, and high muscle mass [9,11].

High lean body mass (LBM), which is the weight of the entire body excluding fat components, reflects an individual’s muscle mass better than BMI and predicts better clinical outcomes for cardiovascular and end-stage renal diseases patients [12,13,14]. However, research on the clinical implications of LBM in postcardiac arrest patients is scarce; hence, we investigated the prognostic impact of LBM on postcardiac arrest patients with the hypothesis that higher LBM is positively associated with better neurological outcomes in cardiac arrest survivors.

## 2. Materials and Methods

### 2.1. Postcardiac Arrest Care and TTM

At our institute, almost 90,000 patients visit the emergency department (ED) each year. Patients with ROSC after out-of-hospital cardiac arrest are assessed and examined in the ED. Patients indicated for TTM receive treatment based on the standardized protocol of our institution. The target temperature for our TTM protocol is 32–36 °C. If a patient has hemodynamic instability, bleeding tendency, or severe infection, a higher target temperature of up to 36 °C could be applied by the treating physician. TTM is conducted with temperature-managing devices with a feedback loop system (Artic Sun^®^ Energy Transfer Pads™; Medivance Corp, Louisville, KY, USA, or Cool Guard Alsius Icy Heat Exchange Catheter; Alsius Corporation, Irvine, CA, USA). All patients receive sedative and analgesic treatments sufficiently. If necessary, their shivering and seizures are also controlled.

### 2.2. Enrolled Patients and Study Design

This retrospective cohort study included adult patients who underwent TTM after out-of-hospital cardiac arrest of nontraumatic origin and were admitted to the ICU for postcardiac arrest care from January 2015 to March 2020 in Ajou University Hospital, Republic of Korea. We excluded patients with end-stage cancer with a life expectancy of less than 6 months, and those who had a do-not-resuscitate order with consent from their families because they had not undergone TTM. In addition, we excluded pediatric patients younger than 18 years, patients who died within 72 h after undergoing TTM or who died owing to sustained multiple organ failure, patients with poor cerebral performance (cerebral performance category (CPC) score of 3 or 4) before cardiac arrest, patients with inadequate temperature management during TTM, and patients with missing or insufficient data. The finally enrolled patients were grouped on the basis of neurological outcomes according to CPC scores. We analyzed and compared the predictive values of BMI and LBM in postcardiac arrest patients.

### 2.3. Data Collection

Study data were retrospectively retrieved from electronic medical records and a data registry of postcardiac arrest patients who underwent TTM after ROSC. Demographic and clinical data, including age, sex, underlying disease (such as hypertension and diabetes), and cardiopulmonary resuscitation (CPR) information (such as whether the cardiac arrest was witnessed, whether bystanders performed CPR, time from collapse to administration of life support, time from the administration of life support until ROSC, first monitored rhythm, whether automated external defibrillator had been used, method of hypothermia, and patient outcomes (expressed as the CPC score)) were collected. Body weight was measured using built-in scales in our ICU beds (HL-SK-154, Hanlim, Cheongju, Korea) at admission. The built-in scales were adjusted to zero before each measurement. Height was also measured at admission as the ratio of the total length of the ICU bed to the height of the patient. We categorized BMI based on classification guidelines of the Korean Society for the Study of Obesity (underweight = BMI <18.5 kg/m^2^; normal weight = BMI: 18.5–22.9 kg/m^2^; overweight = BMI 23.0–24.9 kg/m^2^; obese = BMI ≥25 kg/m^2^) [15].

LBM was calculated using the Kulkarni equation [16].
Men: LBM (kg) = −15.605 − (0.032 × age (y)) + (0.192 × height (cm)) + (0.502 × weight (kg)).
Women: LBM (kg) = −15.034 − (0.018 × age (y)) + (0.165 × height (cm)) + (0.409 × weight (kg))

We classified patients into quartiles according to their calculated LBMs (kilograms) (Level 1 = LBM < 32.71; Level 2 = LBM 32.72–44.17; Level 3 = LBM 44.18–48.97; Level 4 = LBM ≥ 48.98). As the primary outcome of this study, neurological outcome was assessed using the CPC score after 1 month or at discharge if the discharge was sooner than 1 month. A CPC score of 1 (good recovery) to 2 (moderate disability) was categorized as a good neurological outcome, and a CPC score of 3 (severe disability) to 5 (death or brain death) was considered a poor neurological outcome.

This study was conducted based on the principles of the Declaration of Helsinki and was approved by the institutional review board of Ajou University Medical Center (MED-MDB-20-283). The requirement for consent was waived because of the nature of this retrospective study.

### 2.4. Statistical Analysis

Continuous data are presented as mean ± standard deviation or median with interquartile range, and categorical data are presented as numbers with percentages (%). To assess intergroup differences, we used the Mann–Whitney U and Fisher’s exact tests for continuous and categorical variables, respectively. Univariate analysis was performed to identify factors associated with neurological outcomes in postcardiac arrest patients. Univariate and multivariate logistic regression models were used to assess the association between primary outcome and LBM, with LBM considered the categorical variable. We included all possible confounders, such as sex, age, hypertension (yes/no), diabetes (yes/no), bystander basic life support (yes/no), no-flow time (time from collapse to administration of life support (minutes)), total CPR time (time from administration of life support until ROSC (minutes)), ventricular fibrillation or nonperfusion ventricular tachycardia as the first monitored rhythm (yes/no), use of an automatic external defibrillator (AED) during out-of-hospital resuscitation (yes/no), and methods of therapeutic hypothermia (gel pad/intravascular catheter), that could be associated with outcomes. We constructed receiver operating characteristic (ROC) curves to evaluate and compare the predictive values of multivariate models, which included BMI or LBM [17]. Statistical analyses were performed using R software for Mac version 3.2.2 (The R Project, Vienna, Austria). *p*-values <0.05 were considered statistically significant.

## 3. Results

### 3.1. Baseline Characteristics of Patients

Among 512 patients with ROSC after cardiac arrest who were treated in the ICU, 146 patients who did not undergo TTM were excluded. In total, 366 patients underwent TTM, but another 211 patients, including 132 patients who died within 72 h after undergoing TTM or who died owing to sustained multiple organ failure, and 79 patients who had incomplete data, were excluded. Finally, we enrolled 155 patients and categorized them into good- (*n* = 70) and poor neurological outcome groups (*n* = 85) (Figure 1). Baseline characteristics of all enrolled patients are shown in Table 1. Age was significantly different between the two outcome groups. Other general demographics, including sex and underlying disease, did not significantly differ between the two outcome groups. First monitored rhythm, no-flow time, advanced cardiopulmonary life support time, presumed cause of arrest, and the use of AED were statistically different between the two groups (*p* < 0.05). BMI and LBM, as continuous and categorical variables, respectively, were also significantly different between the two groups (*p* < 0.01 and *p* < 0.01, respectively; Table 1).

### 3.2. Analysis of Factors Associated with Neurological Outcomes

In the univariate analysis, high BMI and LBM were associated with good neurological outcomes after 1 month (obese BMI ≥25.00 kg/m^2^: odds ratio (OR), 3.31, 95% CI, 1.50–7.62; *p* = 0.04 and LBM Level 4: OR, 7.18; 95% CI, 2.63–22.17; *p* < 0.01). Furthermore, in the univariate analysis, other factors such as first monitored rhythm, no-flow time, low-flow time, presumed cause of cardiac arrest, and use of AED were associated with neurological outcomes after 1 month (*p* < 0.05). Even after adjusting for multiple confounding factors, including age, sex, history of diabetes, hypertension, presence of a witness, bystander CPR, first monitored rhythm, no-flow time, low-flow time, presumed cause of arrest, use of AED, and the methods of therapeutic hypothermia, LBM Level 4 was still associated with good neurological outcomes (OR, 4.81; 95% CI, 1.10–25.55; *p* = 0.04; Table 2).

### 3.3. Prediction of Neurological Outcomes

Initial high BMI and LBM Level 4 had good associations with good neurological outcomes. DeLong’s test was performed for two areas under the ROC curve (AUROC) of multivariate models which include BMI or LBM. Both two multivariate models include variables representing significant odd ratios in univariate analysis, such as age, first monitored rhythm, total CPR time, no-flow time, and AED use. Additionally, BMI is included in Model 1 and LBM in model 2, respectively. And the two models were not significantly different (0.917 vs. 0.918, *p* value = 0.77; Figure 2).

## 4. Discussion

In this study, a high LBM of patients who had undergone TTM after cardiac arrest was significantly associated with a good neurological prognosis. These results are in line with those of a previous study that reported on the “obesity paradox” using BMI. Recent studies have reported that factors linked to obesity, such as cholesterol level, affect the prognosis of postcardiac arrest patients [18]. However, the mechanism underlying the obesity paradox is not clearly known. Our study demonstrated a relationship between LBM and neurological outcomes in postcardiac arrest patients.

Our result show that LBM and BMI models have similar predictive performance for predicting a good neurologic outcome (Figure 2). LBM is the weight of the entire body excluding fat components and reflects an individual’s muscle mass better than BMI. In other words, patients with high LBM have large skeletal muscle mass, and this may have a favorable effect on postcardiac arrest via several mechanisms (Figure 3). Large muscle mass is associated with a good premorbid condition [19]. Skeletal muscle mass also plays an essential role in maintaining a metabolically healthy phenotype, especially in nonobese individuals. Besides the precardiac arrest condition, postcardiac arrest physiological reserve is better in patients with high LBM than in those with low LBM [20].

Moreover, the in vivo and in vitro neuroprotective potential of the skeletal muscles has been reported as described below. First, a good neurological prognosis after cardiac arrest in patients with high LBM may be due to the large amount of creatine stored in the skeletal muscles. Creatine is one of several potential neuroprotectants against neurodegenerative diseases and ischemic diseases in the heart and brain [21,22]. Because LBM reflects muscle mass better than BMI does, LBM may also be a feasible prognostic factor. Second, a substantial amount of anti-inflammatory myokines excreted from abundant muscle mass may also play a protective role in patients with high LBM. Myokines are cytokines released by skeletal muscle. One of these myokines is interleukin-6 (IL-6), which has an extensive anti-inflammatory effect [23]. A previous study reported that IL-6 has a favorable prognostic effect on cerebral ischemia and dermatological diseases because it inhibits the production of tumor necrosis factor-α and IL-1, which are classical proinflammatory cytokines [24]. Neural damage in postcardiac arrest syndrome could be caused by unbalanced and aggressive inflammatory reactions after ischemia and reperfusion injuries. Therefore, this anti-inflammatory myokine, IL-6, may alleviate the risk of neurological injury arising from aggressive inflammatory insults after cardiac arrest. Third, oxytocin, which is mainly synthesized by the hypothalamic paraventricular and supraoptic nucleus, is expressed in myogenic cells and may have a protective role in patients with postcardiac arrest syndrome. In a recent experimental study, muscle tissue was considered a source of oxytocin [25]. Oxytocin has anti-inflammatory, antiapoptotic, and antioxidant properties that alleviate pathophysiological changes after ischemia–reperfusion in various human organs, including the bladder and heart. Oxytocin may also have a neuroprotective function in neurons against cerebral ischemia via the attenuation of calpain-1, which is implicated in the cell death process [26,27,28]. Therefore, the neuroprotective effect of LBM in patients with postcardiac arrest syndrome might be because of the increased expression of oxytocin. Moreover, skeletal muscles contribute to the stabilization of proteins (proteostasis) and metabolites in neurodegenerative disorders [29]. Cardiac arrest is a condition wherein the homeostasis of various organ functions, including maintenance of protein level, is affected. Thus, large skeletal muscle mass may help restore homeostasis and reduce further neuronal injury after the initial damage of cardiac arrest.

This study had several limitations. First, we used a retrospective design and conducted the study at a single institution in Korea, with a limited number of patients. Thus, the results might not be generalizable to other ethnic groups. In the future, there may be a need for prospective studies involving larger and more diverse patient samples. Second, considering the baseline characteristics of the enrolled patients, there were significant differences in age, and, although not significant, there were differences in the proportion of sex between the two groups according to neurological outcomes. Confounders related to age and sex could have also influenced our results. Third, this study excluded patients who had died within 72 h after undergoing TTM or who died owing to sustained multiple organ failure, because we intended to check the neuroprotective effect of LBM by excluding patients who had died owing to systemic causes. However, this approach may have caused selection bias. Fourth, LBM could be estimated by more accurate modalities, including magnetic resonance imaging, computed tomography, and dual-energy X-ray absorptiometry [30,31]. Because this study was a retrospective one, we calculated the LBM using predictive equations based on data from medical records. The accuracy of the predictive equations may be low in ICU patients [32]. Furthermore, height, which is a key factor in calculating LBM, was estimated using the ratio of the ICU bed because the patient could not stand. Thus, height could have been less accurate than expected, with a possibility of interpersonal variability. Further studies wherein LBM is measured with accurate height values may be necessary. Finally, we used the Kulkarni equation, which was developed in India, among several other predictive equations. It is unclear whether the Kulkarni equation is the most suitable method for Koreans and Asians in general.

Despite these limitations, we believe that LBM could be a meaningful and feasible prognostic factor because it reflects muscle mass more accurately than BMI.

## 5. Conclusions

The study findings demonstrate that an LBM above 48.98 kg could be associated with a good neurological prognosis in patients who undergo TTM after cardiac arrest. Using LBM as an adjunctive prognostic factor, physicians could more precisely predict the neurological outcomes of postcardiac arrest patients. Prospective research is required wherein LBM is measured using more reliable methods, including magnetic resonance, computed tomography, and dual-energy X-ray absorptiometry.

## Figures and Tables

**Figure 1 jcm-10-00071-f001:**
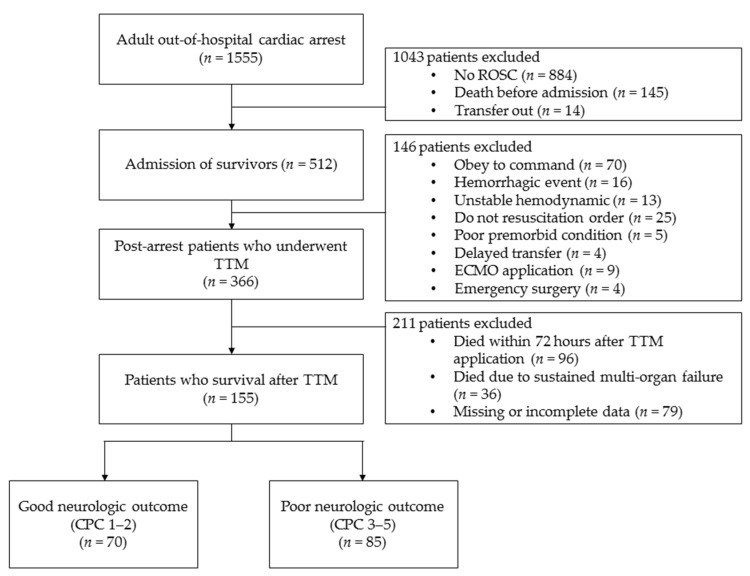
Study flowchart.

**Figure 2 jcm-10-00071-f002:**
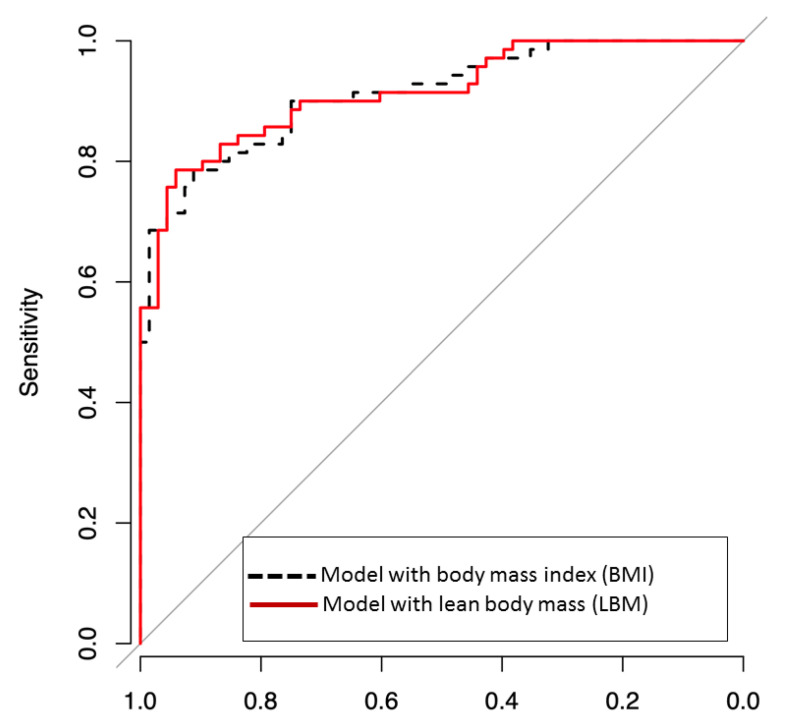
Receiver operating characteristic (ROC) curves for predicting good neurological outcome, defined as cerebral-performance categories 1–2, including body mass index (BMI) or lean body mass (LBM) in multivariate logistic models. Area under ROC curve (AUROC) was a good predictive value for BMI, and LBM predicted good neurological outcomes. The two models were not significantly different (AUROC of BMI vs. AUROC of LBM = 0.917 vs. 0.918, *p*-value = 0.8715).

**Figure 3 jcm-10-00071-f003:**
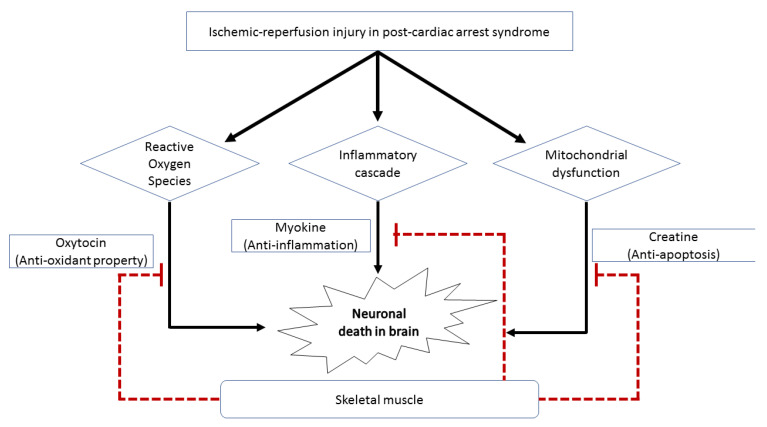
Schematic diagram showing the effects of skeletal muscle on the nervous system. Skeletal muscle can exhibit inhibitory effects on neuronal cell death in survivors after cardiac arrest via the (1) antioxidant property of oxytocin, (2) anti-inflammatory action of myokine, and (3) antiapoptotic activity of creatine.

**Table 1 jcm-10-00071-t001:** Baseline characteristics of the study population.

	Good CPC (*n* = 70)	Poor CPC (*n* = 85)	*p*-Value
Patient characteristics			
Age, (mean) years	50.34 (±13.03)	58.14 (±16.02)	<0.01
Female, *n* (%)	14 (20.00)	29 (34.12)	0.08
Height, (mean) cm	171.30 (±7.59)	165.44 (±8.22)	<0.01
Weight, (mean) kg	70.40 (±13.91)	60.86 (±13.29)	<0.01
Known prearrest health conditions			
Diabetes mellitus, *n* (%)	16 (22.86)	26 (30.59)	0.37
Hypertension, *n* (%)	25 (35.71)	40 (47.06)	0.06
Resuscitation factors			
Witnessed cardiac arrest, *n* (%)	60 (85.71)	65 (76.47)	0.34
Bystander CPR, *n* (%)	50 (71.43)	59 (69.41)	0.58
Initial shockable rhythm, *n* (%)	55 (78.57)	10 (11.76)	<0.01
No-flow time, min (IQR)	2.74 (0.00–4.00)	6.14 (0.00–10.00)	0.02
Total CPR time, mean	20.03 (±12.26)	29.54 (±17.83)	<0.01
Out of hospital BLS time, min	15.37 (±10.95)	20.64 (±12.58)	0.01
ACLS time after hospital arrival, min (IQR)	4.63 (0–4.00)	9.45 (4.00–12.00)	<0.01
Use of AED, *n* (%)	59 (84.29)	23 (27.06)	<0.01
TTM factors			
TTM method, *n* (%)			0.16
Gel pads with feedback loop	62 (87.27)	69 (81.18)	
Intravascular cooling with feedback loop	5 (7.14)	5 (5.88)	
Cooling water blanket	3 (4.29)	11 (12.94)	
Target temperature, *n* (%)			0.70
From 32 to 34 °C	58 (82.86)	74 (87.06)	
From 35 to 37 °C	12 (17.14)	11 (12.94)	
Body-weight-related index			
BMI, mean	24.35 (±4.10)	22.44 (±4.59)	<0.01
Underweight (<18.50), *n* (%)	3 (4.29)	13 (15.29)	<0.01
Normal (18.50–22.99), *n* (%)	25 (35.71)	40 (47.06)	
Overweight (23.00–24.99), *n* (%)	13 (18.57)	18 (21.18)	
Obese (≥25.00), *n* (%)	29 (41.43)	14 (16.47)	
LBM, mean	48.90 (±9.21)	42.64 (±10.74)	<0.01
Level 1 (<32.71), *n* (%)	6 (8.58)	21 (24.71)	<0.01
Level 2 (32.72–44.17), *n* (%)	12 (17.14)	23 (27.06)	
Level 3 (44.18–48.97), *n* (%)	11 (15.71)	21 (24.71)	
Level 4 (≥48.98), *n* (%)	41 (58.57)	20 (23.53)	

CPC = cerebral performance category, CPR = cardiopulmonary resuscitation, BLS = basic life support, ACLS = advanced cardiovascular life support, AED = automatic external defibrillator, TTM = targeted temperature management, BMI = body mass index, LBM = lean body mass.

**Table 2 jcm-10-00071-t002:** Logistic regression models for clinical outcomes. Univariate odds ratios adjusted for multiple confounders.

	Univariate OR (95% CI)	*p*-Value	Multivariate OR (95% CI), Model Including BMI ^e^	*p* Value	Multivariate OR (95% CI), Model Including LBM ^f^	*p*-Value
Age	0.96 (0.94–0.98)	<0.01	0.96 (0.93–0.99)	0.02	0.96 (0.92–0.99)	0.03
Initial shockable rhythm	27.50 (11.97–69.18)	<0.01	18.32 (5.04–79.17)	<0.01	17.38 (4.64–70.27)	<0.01
No-flow time ^a^	0.93 (0.87–0.98)	0.01	0.96 (0.88–1.02)	0.24	0.96 (0.89–1.03)	0.30
Total CPR time ^b^	0.95 (0.93–0.97)	<0.01	0.96 (0.92–0.99)	0.01	0.95 (0.91–0.98)	0.01
Use of AED	13.53 (6.25–31.51)	<0.01	1.39 (0.33–5.27)	0.63	1.59 (0.38–6.03)	0.51
BMI (Cont.)	1.13 (1.04–1.25)	<0.01	1.10 (0.98–1.25)	0.13		
BMI ^c^						
Underweight (<18.50)	0.37 (0.07–1.28)	0.15	0.40 (0.04–3.39)	0.44	-	-
Normal (18.50–22.99)	1	1	1	1	-	-
Overweight (23.00–24.99)	1.16 (0.48–2.76)	0.74	1.96 (0.56–7.12)	0.30	-	-
Obese (≥25.00)	3.31 (1.50–7.62)	<0.01	3.27 (1.04–11.07)	0.04	-	-
LBM (Cont.)	1.07 (1.03–1.11)	<0.01			1.05 (0.99–1.10)	0.09
LBM ^d^						
Level 1 (<32.71)	1	1	-	-	1	1
Level 2 (32.72–44.17)	1.82 (0.60–6.06)	0.30	-	-	1.31 (0.25–7.37)	0.75
Level 3 (44.18–48.97)	1.83 (0.59–6.18)	0.31	-	-	9.85 (0.18–5.52)	0.98
Level 4 (≥48.98)	7.18 (2.63–22.17)	<0.01	-	-	4.81 (1.10–25.55)	0.04

CPR = cardiopulmonary resuscitation, AED = automatic external defibrillator, BMI = body mass index, LBM = lean body mass, Cont. = as continuous variable.^a^ No-flow time: time from collapse to administration of resuscitation^b^ Total CPR time: time from administration of resuscitation to return of spontaneous circulation (BLS time + ACLS time)^c^ Normal body mass index group as reference group.^d^ Lowest lean body mass group as reference group.^e^ Hosmer–Lemeshow X squared = 6.95, df = 8, *p*-value = 0.541, indicating good model fit. Hosmer and Lemeshow test (binary model).^f^ Hosmer–Lemeshow X squared = 9.77, df = 8, *p*-value = 0.281, indicating good model fit. Hosmer and Lemeshow test (binary model).

## Data Availability

Please contact author for data requests.

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
