# Peer review of "Predictive Value of Estimated Lean Body Mass for Neurological Outcomes after Out-of-Hospital Cardiac Arrestâ€"

_jcm, 2020, doi:10.3390/jcm10010071_

Round 1
Reviewer 1 Report
The title of the manuscript, titled:” Predictive value of estimated lean body mass for the neurological outcome after out-of-hospital cardiac arrest”, is very eye-catching, and it makes readers want to look for something like continuing to read such as providing doctors with important clinical diagnostic indicators. Unfortunately, reader may be disappointed after reading thoroughly. In particular, significant differences in age lead to be difficult to judge the causes of other parameters differences because many diseases are aging dependent. I have suggestion that author consider to avoid aging dependent and recalculate the data to see what will be shown. In addition, 1) it’s checked for plagiarism by ithenticate tool and found the manuscript has 46% plagiarism, please write your manuscript using your own word; 2) the manuscript discussed possible factors, involved in mechanisms such as IL-6, Oxytocin, Calpain-1 and creatinine, but there is no data for these factors in the results ‘part. It is very easy to detect these factors. I hope to provide the data on some of these factors to support the discussion and provide useful materials for further related basic research and clinical applications. Otherwise, the manuscript has defects either in the theory or in the application significance; 3) All abbreviated should be defined to be more understanding.
Reviewer 2 Report
The purpose of the present study was to investigate the prognostic impact of LBM on post-cardiac arrest patients. Authors concluded that initial high LBM is a feasible prognostic factor for good neurological outcome in post-cardiac arrest patients. This study was written well, however, this study need to be revised for several points.
1) Introduction:
What is the hypothesis of this study? Please describe in INTRODUCTION.
2)Authos used U test and Fisher’s exact tests for the analysis. So, please add the value of the test statistic in Table 1.
3) Is there anything else about the limits of research?
For example, gender difference etc.
Reviewer 3 Report
JCM_leanbodymass_review-02.12.2020
In a retrospective cohort study including adult patients surviving outdoor cardiac arrest between 2015 and 2020 the relationship between estimated lean body mass (LBM) and prognosis of these patients was evaluated.
Non neurological causes of death were excluded, and the enrolled patients were divided in good and in poor outcome groups as estimated by the “cerebral performance category score” (CPC). The estimated LBM was categorized by quartiles. Multivariate regression analysis was used to evaluate the association between LBM and CPC score (“cerebral performance category score”). N = 155 patients were analysed by comparing CPC score 1-2 (n = 70) versus 3-5 (n = 85). High LBM was associated with a good prognosis with respect to neurological outcomes in these post-cardiac arrest patients.
Comments:
- The presented results are of interest as well as the discussion of the potential neuroprotective potential of the skeletal muscle. The results underscore the urgent need of regular and continuous exercise training especially in the elderly population. I would recommend to summarise these potential biochemical interactions in diagram including a figure legend referring to the scientific literature of all these potential protective mechanisms. This would help the reader to percept and understand these mechanisms.
- The population under investigation is highly selective by only representing 10% of the original population. This selection is needed as the study is focussing on neurological interactions and potentially protective mechanisms. However, we have to be aware that tis selection does not necessarily reflect all day care, as the majority of patients with out-of-hospital cardiac arrest also may primarily suffer from cardiovascular disease leading to threatening cardiac arrhythmias. Depending on the time period of low flow circulation (being influenced by a variety of “external” and “internal” pre-conditions) transient and persistent neurological damage varies, and I am not sure, whether these preconditions markedly influencing the general clinical outcome can be calculated out.
- The definition of “neurological death” is unclear. Neurological death per se means “brain death” whereas the function of the cardiovascular system and of other vital organs is still ensured. How has brain death beet measured and witnessed? There also may be conditions that may not really allow a clear differentiation between brain death and cardiovascular death.
- Occurrence of brain death may markedly be influenced by severely altered function of the circulation, thereby it may be difficult to distinguish between the primary causes of death.
- It is within the nature of such studies that not all potential confounders can be included into multivariate regression analysis. But some potential confounders may severely interfere with the results. These confounders include pre-existing atherosclerosis, heart failure, smoking, being treated for cardiovascular prevention yes or no etc. The authors should comment on this.
Minor comments:
- The abbreviations should be separately outlined
- Some polishing of the English language
Round 2
Reviewer 3 Report
Dear authors,
I acknowledge the effort in improving the English language throughout the manuscript. Still I strongly recommend language revision by a native English speaker or a professional translater.
Fig. 1, "study flow chart" has been presented twice, please correct
The addditional Fig. 3 is fine.
In conclusion, you have addressed the reviewer`s objections, but a profound English language editing still is mandatory before this manuscript can be recommended for publication. This also is in the well respected interest of the authors.
Author Response
Thank your for your kind comments.
We had got a language editing service provided by MDPI journal was performed. (The certificate file is attached) And, after giving us the comment, we received additional language editing service provided by professional editing company (Editage by CACTUS). We revised the manuscript using the "Track Changes" function in MS word.
And, we checked that Figure 1 is duplicated, and we deleted one.
Thank you.
